# Modulation of Canine Melanocortin-3 and -4 Receptors by Melanocortin-2 Receptor Accessory Protein 1 and 2

**DOI:** 10.3390/biom12111608

**Published:** 2022-11-01

**Authors:** Ren-Lei Ji, Shan-Shan Jiang, Ya-Xiong Tao

**Affiliations:** Department of Anatomy, Physiology, and Pharmacology, College of Veterinary Medicine, Auburn University, Auburn, AL 36849, USA

**Keywords:** canine, melanocortin-2 receptor accessory protein, melanocortin-3 receptor, melanocortin-4 receptor, pharmacology, splice variant

## Abstract

The neural melanocortin receptors (MCRs), melanocortin-3 and -4 receptors (MC3R and MC4R), have crucial roles in regulating energy homeostasis. The melanocortin-2 receptor accessory proteins (MRAPs, MRAP1 and MRAP2) have been shown to regulate neural MCRs in a species-specific manner. The potential effects of MRAP1 and MRAP2 on canine neural MCRs have not been investigated before. Herein, we cloned canine (c) *MC3R* and identified one canine *MRAP2* splice variant, *MRAP2b*, with N-terminal extension of cMRAP2a. Canine MC3R showed higher maximal responses to five agonists than those of human MC3R. We further investigated the modulation of cMRAP1, cMRAP2a, and cMRAP2b, on cMC3R and cMC4R pharmacology. For the cMC3R, all MRAPs had no effect on trafficking; cMRAP1 significantly decreased B_max_ whereas cMRAP2a and cMRAP2b significantly increased B_max_. Both MRAP1 and MRAP2a decreased R_max_s in response to α-MSH and ACTH; MRAP2b only decreased α-MSH-stimulated cAMP generation. For the MC4R, MRAP1 and MRAP2a increased cell surface expression, and MRAP1 and MRAP2a increased B_max_s. All MRAPs had increased affinities to α-MSH and ACTH. MRAP2a increased ACTH-induced cAMP levels, whereas MRAP2b decreased α-MSH- and ACTH-stimulated cAMP production. These findings may lead to a better understanding of the regulation of neural MCRs by MRAP1 and MRAP2s.

## 1. Introduction

Melanocortin-3 and -4 receptors (MC3R and MC4R), also called neural MCRs, are two melanocortin receptors (MCRs) that are highly expressed in the central nervous system [1,2,3,4], and are essential for regulating energy homeostasis [5,6,7]. Mice lacking *Mc3r* have a moderate obesity phenotype with normal food intake and metabolism, decreased lean mass and increased fat mass [8,9,10]. *Mc4r* knockout mice have increased food intake, decreased energy expenditure, and morbid obesity [11,12]. These findings suggest a distinct nonredundant regulation of the energy balance by MC3R and MC4R. Additionally, mutations in *MC3R* and *MC4R* are associated with obesity [7,13,14,15,16]. MC3R was shown to be involved in other physiological functions, including the modulation of natriuresis [17], cardiovascular function [18,19], immune response [20,21,22,23,24], and timing of sexual maturation [25]. MC4R also has crucial roles in sexual function and reproduction [7,15].

MC3R and MC4R are activated by the endogenous agonists, including α-, β-, γ-melanocyte-stimulating hormones (MSHs) and adrenocorticotropin (ACTH) [26,27]. As members of the family of G protein-coupled receptors (GPCRs), the neural MCRs primarily couple to the stimulatory G protein (Gαs) to stimulate adenylyl cyclase activity, resulting in increased generation of the intracellular second messenger cyclic adenosine monophosphate (cAMP) to trigger downstream signaling.

MCRs have been shown to interact with small single transmembrane proteins, melanocortin-2 receptor accessory proteins (MRAPs, including MRAP1 and MRAP2) [28,29,30,31] (reviewed in [32,33]). MRAP1 was first identified as the specific chaperone for MC2R, essential for MC2R forward trafficking [28,29,33,34]. Human (h) *MRAP1* mutations account for ~20% of familial glucocorticoid deficiency cases [28,35]. Subsequent studies suggest that MRAP1 possesses functions beyond regulating MC2R. Indeed, hMRAP1 has been shown to modulate five human MCRs in distinct ways [30,36,37]. MRAP1 regulates MC3R and MC4R in chicken and frog [25,38,39]. So far, there are only a few studies focusing on MRAP1 regulation of neural MCRs [25,30,37,38,39,40,41,42,43].

MRAP2, a paralog of MRAP1, with high expression in the brain, has essential roles in regulating energy homeostasis. Mice lacking *Mrap2* show early onset severe obesity [44,45]. Human *MRAP2* mutations are also associated with severe obesity [44,46,47,48]. MRAP2 has been shown to modulate MC3R/MC4R trafficking and signaling in vertebrates [30,31,38,43,44,49,50,51,52,53]. Human *MRAP2* has three alternatively spliced variants with different C-termini in the proteins, and these three variants exert different effects on MC3R/MC4R pharmacology [43].

From the studies on the regulation of MC3R and MC4R by MRAPs, it was shown that MRAPs modulate neural MCRs in a receptor- and species-dependent manner. The effects of MRAP1 and MRAP2 on canine (*Canis lupus familiaris*) (c) MC3R and cMC4R have not been studied before. Herein, we investigated the pharmacology of cMC3R as well as the pharmacological modulation of cMC3R and cMC4R by cMRAP1 and cMRAP2a/cMRAP2b. We have previously reported the pharmacology of cMC4R [54].

## 2. Materials and Methods

### 2.1. Ligands and Plasmids

[Nle^4^,D-Phe^7^]-α-MSH (NDP-MSH) and D-Trp^8^-γ-MSH were obtained from Vivitide (Louisville, KY, USA). Human α-MSH and β-MSH were purchased from Pi Proteomics (Huntsville, AL, USA). Human ACTH (1-24) was supplied by Phoenix Pharmaceuticals (Burlingame, CA, USA). Canine α-MSH, β-MSH, and ACTH (1-24), share the same sequences with the corresponding human counterparts. [^125^I]-cAMP and [^125^I]-NDP-MSH were iodinated using chloramine T method [55,56]. N-terminal myc-tagged hMC3R, cMC3R, and cMC4R [54]) and N-terminal Flag-tagged MRAPs (cMRAP1, cMRAP2a, and cMRAP2b) were commercially synthesized and subcloned into pcDNA3.1 by GenScript (Piscataway, NJ, USA).

### 2.2. Cell Culture and Transfection

Human embryonic kidney (HEK) 293T cells (ATCC, Manassas, VA, USA) were cultured at 37 °C in a 5% CO_2_-humidified incubator [57]. Cells were plated into gelatin-coated 6- or 24-well plates. Cells were co-transfected with 0.25 μg/μL cMC3R or cMC4R with or without MRAP plasmids using calcium phosphate precipitation method [58].

### 2.3. Ligand Binding Assays

Binding assay was performed as described previously [49,57]. The ligands and their final concentrations used in this study were NDP-MSH (from 10^−12^ to 10^−6^ M), α-MSH (from 10^−11^ to 10^−5^ M), β-MSH (from 10^−11^ to 10^−5^ M), ACTH (1-24) (from 10^−12^ to 10^−6^ M), and D-Trp^8^-γ-MSH (from 10^−12^ to 10^−6^ M). To investigate the regulation of MRAPs on the binding properties of cMC3R and cMC4R, cMC3R or cMC4R (0.25 μg/μL) and cMRAP1, cMRAP2a or cMRAP2b plasmids in two ratios (1:0 and 1:5) were co-transfected into 6-well plates, and two ligands, α-MSH and ACTH (1-24), were used for binding and signaling assays.

### 2.4. Ligand-Stimulated cAMP Assays

Radioimmunoassay (RIA) was used to determine intracellular cAMP levels as described previously [55,57]. Five ligands, including NDP-MSH, α-MSH, β-MSH, ACTH (1-24), and D-Trp^8^-γ-MSH, were used. To investigate effects of MRAPs on cMC3R or cMC4R signaling, cells were co-transfected with cMC3R or cMC4R (0.25 μg/μL) and cMRAP plasmids (1:5), and two ligands, α-MSH and ACTH (1-24), were used.

### 2.5. Flow Cytometry Assay

The regulation of cMRAP1 or cMRAP2s on the expression of cMC3R and cMC4R was carried out using flow cytometry (Accuri Cytometers, Ann Arbor, MI, USA) as described previously [59,60]. Cells were co-transfected with cMC3R or cMC4R and cMRAP1, cMRAP2a, or cMRAP2b plasmids (1:5). Fluorescence of cells transfected with pcDNA3.1 was used for background staining. The expression of receptors was calculated as the percentage of the cell transfected with receptors in absence of MRAPs (set as 100%) [59].

### 2.6. Statistical Analysis

All data were presented as mean ± S.E.M. GraphPad Prism 8.3 software (GraphPad, San Diego, CA, USA) was used to calculate the parameters of ligand binding, cAMP signaling, and flow cytometry assay. The significance in binding and signaling parameters between cMC3R and hMC3R, as well as vehicle and ligand-treated groups, were all determined by Student’s *t*-test. One-way ANOVA was used to analyze the significant differences in binding, cAMP, and flow cytometry between multiple groups.

## 3. Results

### 3.1. Nucleotide and Deduced Amino Acid Sequences of cMC3R, cMRAP1, and cMRAP2s

The canine *MC3R* (GenBank: NM_001135124.1) had 972 bp open reading frame (ORF), encoding a putative protein of 323 amino acids with 35.79 kDa molecular mass (Figure 1A). cMC3R had seven hydrophobic transmembrane domains (TMDs). Several conserved motifs, including PMY, DRY, and DPxxY, and three potential *N*-linked glycosylation sites (Asn^2^, Asn^16^, and Asn^28^) in N-terminus, were present at homologous positions with MC3Rs of other species (Figure 1). Canine MC3R shared high identities with other MC3R orthologs, 96% to giant panda, 92% to human, 90% to cat, 89% to pig, 87% to mouse, 76% to chicken, 76% to turtle, 75% to frog, and 72% to zebrafish. The phylogenetic tree showed that cMC3R nested with mammalian MC3Rs (Figure 1B).

The canine *MRAP1* had 336 bp ORF that encoded a putative protein of 111 amino acids with 12.90 kDa molecular mass (Figure 2A). cMRAP1 had the classical characteristic of other MRAP orthologs, including two potential *N*-linked glycosylation sites (Asn^3^ and Asn^6^), YEYY motif, LDYL motif, LKANKYL motif, and a single TMD (Figure 2A). cMRAP1 shared high identities with cat MRAP1 (90%), and lower identities with other MRAP1 (44–79%). The phylogenetic tree showed that cMRAP1 clustered with mammalian MRAP1s and was evolutionarily closer to red fox MRAP1 (Figure 2B).

The canine *MRAP2* consists of 11 exons. Two *MRAP2* splice variants were identified: *MRAP2a* (XM_038682814.1) derived from four exons (2, 6, 7, and 11) that had 621 bp ORF, encoding a putative protein of 206 amino acids with 23.62 kDa molecular mass (Figure 3A); *MRAP2b* (XM_038682813.1) derived from five exons (3, 4, 6, 7, and 11) that had 684 bp ORF encoding a putative protein of 232 amino acids with 26.71 kDa molecular mass (Figure 3A). Canine MRAP2b had an extended N-terminus (26 amino acids) compared with cMRAP2a, and they shared the common structure with other MRAP2s, such as one potential *N*-linked glycosylation sites (Asn^9^ in MRAP2a and Asn^35^ in MRAP2b), YEYY motif, LKAHKYS motif, and a single TMD (Figure 3B,C). Multiple sequence alignment analysis showed that dog MRAP2a and MRAP2b shared high identities with mammalian MRAP2s (>81%) and lower identities with MRAP2s from other species (<78%). The two isoforms of cMRAP2 were clustered with different MRAP2s, in which MRAP2a was evolutionarily closer to cat MRAP2, and MRAP2b was nested with Nile tilapia MRAP2 (Figure 3D).

### 3.2. Ligand Binding Properties of cMC3R

The binding assay was performed using multiple MC3R ligands, including NDP-MSH, α-MSH, β-MSH, ACTH, and D-Trp^8^-γ-MSH. We included hMC3R for comparison in the same experiments to explore whether cMC3R shows any unique pharmacological characteristics. The maximal binding value (B_max_) of cMC3R was 247.97 ± 13.44% of that of hMC3R (Figure 4 and Table 1). Canine MC3R had significantly lower affinities to NDP-MSH and D-Trp^8^-γ-MSH than that of hMC3R (Figure 4 and Table 1). The two MC3Rs showed similar IC_50_s when α-MSH, β-MSH, and ACTH, were used (Figure 4 and Table 1).

### 3.3. cAMP Signaling Properties of cMC3R

Intracellular cAMP levels were determined to explore whether cMC3R could respond to these agonists. All agonists could dose-dependently stimulate cMC3R and increase cAMP production (Figure 5 and Table 2). Canine MC3R showed higher maximal responses (R_max_) in response to four agonists (NDP-MSH, β-MSH, ACTH, and D-Trp^8^-γ-MSH) compared to hMC3R (Figure 5 and Table 2). Similar EC_50_s between the two MC3Rs were observed in response to five agonists (Figure 5 and Table 2). Additionally, cMC3R showed similar basal activity as hMC3R (Table 2).

### 3.4. Modulation of cMC3R and cMC4R Expression by MRAPs

Canine MC3R or MC4R expression regulated by MRAPs was measured using flow cytometry. Results showed that cMRAP1, cMRAP2a, and cMRAP2b had no effect on the cell surface and total expression of cMC3R (Figure 6A,B). For cMC4R, cMRAP1 and cMRAP2a increased cell surface and total expression, and cMMRAP2b only increased total expression of cMC4R (Figure 6C,D).

### 3.5. Modulation of cMC3R Pharmacology by MRAPs

Ligand binding assays with α-MSH and ACTH showed that cMRAP1 decreased the B_max_, whereas cMRAP2a and cMRAP2b increased B_max_s of cMC3R (Figure 7A,B and Table 3). All MRAPs had no significant effect on IC_50_s of cMC3R to α-MSH and ACTH (Figure 7A,B and Table 3). Modulation of cMRAPs on cMC3R signaling was also studied. Results showed that all MRAPs did not alter α-MSH and ACTH potencies of cMC3R (Table 4). Both MRAP1 and MRAP2a decreased R_max_s in response to α-MSH and ACTH; MRAP2b significantly decreased α-MSH-stimulated cAMP generation but showed similar ACTH-induced cAMP production of cMC3R (Figure 7C,D and Table 4). Only MRAP2b decreased the basal activity, and the other MRAPs had no effect on the basal cAMP levels of cMC3R (Table 4).

### 3.6. Modulation of cMC4R Pharmacology by MRAPs

Ligand binding assays showed that MRAP1 and MRAP2a increased B_max_s of cMC4R and MRAP2b did not affect the B_max_ (Figure 8A,B and Table 5). All MRAPs increased the affinities of cMC4R to α-MSH and ACTH (Figure 8A,B and Table 5).

Signaling results showed that all MRAPs did not affect EC_50_s of cMC4R in response to α-MSH and ACTH (Table 6). MRAP1 had no effect on α-MSH- and ACTH-stimulated cAMP level of cMC4R (Figure 8C,D and Table 6). MRAP2a decreased α-MSH- and ACTH-induced cAMP signaling, whereas MRAP2b increased ACTH-stimulated signaling and did not affect α-MSH-stimulated cAMP signaling (Figure 8C,D and Table 6). Canine MC4R showed higher basal cAMP production than that of hMC4R (2.38 times that of hMC4R), indicating that cMC4R might be constitutively active. In this study, all MRAPs decreased the basal cAMP levels of cMC4R (Table 6).

## 4. Discussion

In this study, we cloned canine *MC3R* and investigated its pharmacological properties. We have reported cMC4R pharmacology previously [54,61]. In the current study, we also identified a *MRAP2* variant, *MRAP2b*. The potential regulation of cMRAP1 and two cMRAP2 isoforms on cMC3R/cMC4R pharmacology were further studied.

To investigate the pharmacology of cMC3R, ligand binding and signaling assays were performed. cMC3Rs showed a higher binding capacity than that of hMC3R, consistent with the results of other MC3Rs, including channel catfish [49], topmouth culter [52], and giant panda [62]. For signaling, different from the results of giant panda and pig MC3Rs [62,63], cMC3R had a higher ligand-induced cAMP level than that of hMC3R. In addition, our current results are consistent with mammalian MC3Rs in that MC3R has little or no basal cAMP signaling [62,63,64,65]. Of interest, high constitutive activities were present in several non-mammalian MC3Rs, including teleosts [49,52,66], amphibians (Mexican axolotl) [67], and avian (chicken) [38,68]. The amino acids accounting for the differences in the constitutive activity between these MC3Rs are not clear. N-termini and extracellular loops are essential for the modulation of constitutive activities in hMC4R [69,70], luteinizing hormone receptor [71], and thyroid-stimulating hormone receptor [72,73]. Lower homology is observed in the N-termini and extracellular loops of mammalian and non-mammalian MC3Rs. Further studies are needed to determine the exact molecular determinants.

The potential roles of cMRAPs on cMC3R/cMC4R trafficking were studied. Human MRAP1a decreased [30,74] or increased [43], and MRAP2a decreased [30,74] or had no effect [43] on the cell surface expression of hMC3R. The current study showed that all cMRAPs did not affect the cell surface expression of cMC3R. In other species, frog MRAP1 increased and chicken MRAP1 did not alter the surface expression of MC3Rs [38,39]. MRAP2 decreased the surface expression of clawed frog MC3R [39], increased the surface expression of topmouth culter Mc3r [52], and had no effect on the surface expression of Mexican axolotl and chicken MC3Rs [38,67]. For MC4R, hMRAP1a and hMRAP2a decreased [30,74] or increased [43] the cell surface expression of hMC4R. In this study, cMRAP1 and cMRAP2a increased cell surface expression of cMC4R. MRAP1 had no effect on the surface expression of chicken MC3R [38] and increased frog MC3R expression [42]. MRAP2 has been reported to decrease the surface expression of tilapia and Mexican axolotl MC4Rs [67,75], increase the membrane expression of zebrafish (Mrap2b) [31], topmouth culter (Mrap2a and Mrap2b) [51], and *Xenopus* MC4Rs [39], or have no effect on the surface expression of chicken and snakehead MC4Rs [38,53]. Collectively, MRAP1 and MRAP2 modulate the MC3R/MC4R trafficking to the plasma membrane in a species- and receptor-specific manner.

Pharmacological studies were further performed on the potential MRAP modulation of cMC3R. Human MRAP1 and MRAP2 decreased α-MSH- and ACTH-induced [43] or increased α-MSH-stimulated [37,74] cAMP production of hMC3R. *Xenopus* MRAP1 increased α-MSH- and ACTH-induced cAMP signaling, and chicken MRAP1 did not affect agonist-induced signaling of MC3Rs [38,39]. Mrap2-decreased Mc3r signaling was also reported in channel catfish, topmouth culter (Mrap2a) [51], and Mexican axolotl [67], whereas MRAP2-increased MC3R signaling was observed in chicken and *Xenopus* MC3Rs [39]. Zebrafish Mrap2s did not affect agonist-induced signaling of MC3R [31]. Our results showed that MRAP1 did not alter MC3R trafficking, decreased B_max_, and α-MSH- and ACTH-induced cAMP levels, indicating that interaction between cMRAP1 and cMC3R might inhibit cMC3R bound to ligands, resulting in decreased signaling; MRAP2s did not change cMC3R trafficking, increased B_max_s, but decreased signaling, probably due to the interaction of cMRAP2s and cMC3R leading to conformation change, further inhibiting G protein-induced intracellular cAMP signaling.

The potential modulation of cMRAPs on cMC4R pharmacology was also studied. Conflicting results were reported previously on hMRAP1a- and hMRAP2a-regulated signaling of hMC4R, where hMRAP1a decreased α-MSH-induced or did not affect α-MSH- and ACTH-stimulated hMC4R signaling [37,43]; MRAP2a had no effect [76], decreased [43] or increased [41,74] α-MSH-stimulated and did not affect ACTH-induced [43,76] signaling of hMC4R. Chicken MRAP1 was shown to decrease α-MSH-stimulated and have no effect on ACTH-induced signaling of MC4R [38]. MRAP1 increased α-MSH- and ACTH-stimulated signaling of *Xenopus* MC4R [39]. MRAP2-suppressed α-MSH- and/or ACTH-stimulated signaling of MC4Rs were present in several teleosts [31,51,53,75,77]. Our results indicated that MRAP1 had no effect on the efficacy of cMC4R in response to α-MSH and ACTH, MRAP2a increased ACTH-stimulated but had no effect on α-MSH-induced cMC4R signaling, and MRAP2b decreased α-MSH- and ACTH-induced signaling, suggesting that MRAP1 modulated cMC4R trafficking and ligand binding, but did not affect signaling; the interaction of MRAP2a and cMC4R might change MC4R ligand selectivity and sensitivity; MRAP2b might inhibit receptor coupling to G protein, resulting in decreased signaling. Hence, MRAPs might be involved in regulating receptor ligand selectivity and sensitivity in a species-dependent manner. The potential mechanisms of the MRAP regulation of receptor ligand selectivity and sensitivity need further study.

Human MC4R shows modest basal cAMP signaling [65]. The defect in basal activities of *MC4R* mutations can cause obesity [69,78]. Mrap2- and Agrp-suppressed basal activity of Mc4r play an important role in promoting the growth of zebrafish and culter [31,52,79]. These studies indicate that the basal activity of MC4R plays a pivotal role in the modulation of energy homeostasis [80]. Human MRAP1s increased [37,41,43] or did not affect [30,36] the basal activities of hMC4R, and MRAP2s decreased [43] or had no effect [30,36,41,81] on the basal activities of hMC4R. Our studies showed that cMRAP1 and cMRAP2s decreased the basal cAMP signaling of cMC4R. Decreased MC4R basal activities by MRAP2(s) were also reported in other species [31,51,53,75,77].

Alternative splicing is prevalent in eukaryotes and isoforms generated by alternative splicing might have different functions [82,83,84,85]. Splicing variants provide a nature-made chance to investigate the roles of specific domains. Human *MRAP* and *MRAP2* have two and three alternatively spliced forms, respectively, and they show different effects on the hMC3R/hMC4R pharmacology [28,43]. Canine *MRAP2* also had two alternatively spliced variants, *MRAP2a* and *MRAP2b* (MRAP2b with extension at N-terminus compared with MRAP2a). This extension sequence at the N-terminus of MRAP2b is not found in other MRAP2s. Our results showed that MRAP2a and MRAP2b had different effects on cMC3R/cMC4R pharmacology. The N-termini of MRAP1 and MRAP2, with several conserved motifs, have important roles in modulating GPCR pharmacology [43,86,87,88]. We speculate that the extension sequences at the N-termini of MRAP2 might have important roles in MC3R/MC4R pharmacology.

## 5. Conclusions

In summary, we cloned canine *MC3R* and investigated its pharmacology, as well as modulation of MC3R and MC4R pharmacology by MRAPs. MRAP1 did not affect the MC3R trafficking and decreased α-MSH- and ACTH-induced signaling, whereas MRAP1 increased the cell surface expression and decreased the basal activity of cMC4R. The two MRAP2 isoforms exerted different effects on cMC3R or MC4R pharmacology. MRAP2a decreased α-MSH- and ACTH-induced signaling, whereas MRAP2b only the decreased α-MSH-stimulated signaling of cMC3R. MRAP2a increased the cell surface expression and ACTH-induced signaling, decreased the basal activity of cMC4R, whereas MRAP2b had no effect on trafficking, and decreased basal and α-MSH- and ACTH-induced signaling. This study contributes to a better understanding of cMC3R/cMC4R.

## Figures and Tables

**Figure 1 biomolecules-12-01608-f001:**
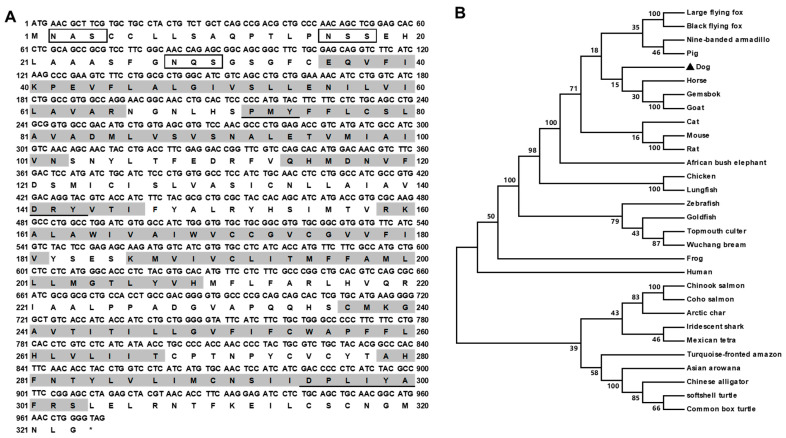
Nucleotide and deduced amino acid sequences (**A**) and phylogenetic tree (**B**) of cMC3R. Positions of nucleotide and amino acid sequences are indicated on both sides. N-linked glycosylation sites are present in open boxes. Shaded boxes show putative TMD1-7. The conserved motifs (PMY, DRY and DPxxY) are underlined. Asterisk (*) shows stop codon. The tree was constructed by the neighbor-joining (NJ) method. Numbers at nodes indicate the bootstrap value, as percentages, obtained for 1000 replicates. Black dot denotes canine MC3R. MC3Rs: *Canis lupus familiaris* (dog, NM_001135124.1), *Capra hircus* (goat, XP_005688382.1), *Xenopus tropicalis* (frog, XP_002935436.1), *Culter alburnus* (topmouth culter, MT419813), *Carassius auratus* (goldfish, BAJ83473.1), *Danio rerio* (zebrafish, AAO24744.1), *Homo sapiens* (human, NP_063941.3), *Sus scrofa* (pig, AFK25142.1), *Mus musculus* (mouse, AAI03670.1), *Gallus gallus* (chicken, XP_004947293.1), *Rattus norvegicus* (rat, NP_001020441.3), *Pangasianodon hypophthalmus* (iridescent shark, XP_026770221.1), *Equus caballus* (horse, NP_001243901.1), *Pteropus vampyrus* (large flying fox, XP_011368476.1), *Pteropus alecto* (black flying fox, XP_006921991.1), *Felis catus* (cat, XP_023106851.1), *Loxodonta africana* (African bush elephant, XP_003419952.1), *Salvelinus alpinus* (Arctic char, XP_023994975.1), *Pelodiscus sinensis* (Chinese softshell turtle, XP_006129463.1), and *Alligator sinensis* (Chinese alligator, XP_006018246.1).

**Figure 2 biomolecules-12-01608-f002:**
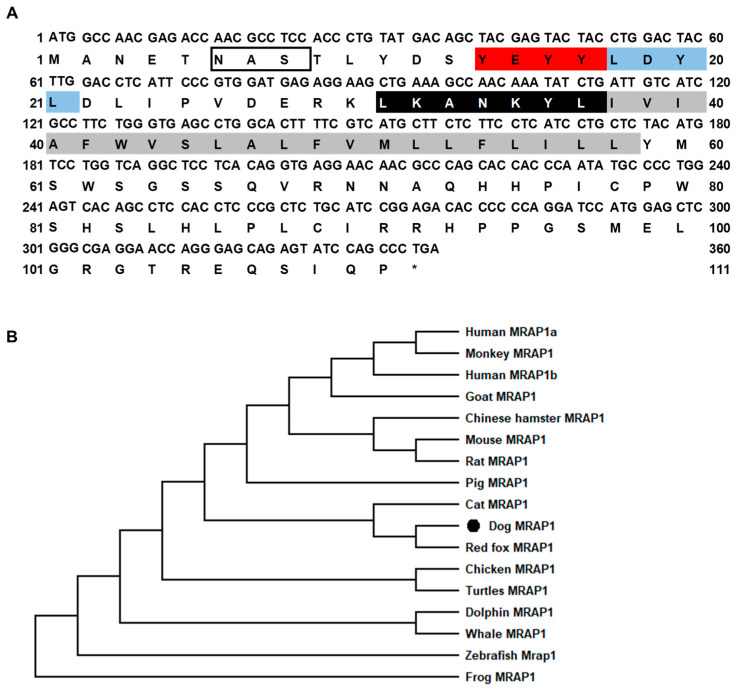
Nucleotide and deduced amino acid sequences (**A**) and phylogenetic tree (**B**) of cMRAP1. Positions of nucleotide and amino acid sequences are indicated on both sides. *N*-linked glycosylation sites are present in open boxes. Red box denotes YEYY motif. Light blue box indicates LDYL motif. Black box is LKANKYL motif. Shaded box shows putative TMD. Asterisk (*) shows stop codon. The tree was constructed by the neighbor-joining (NJ) method. Numbers at nodes indicate the bootstrap value, as percentages, obtained for 1000 replicates. Black dot denotes canine MRAP1. MRAP1s: *Canis lupus familiaris* (dog, XP_005638887.1), *Homo sapiens* (human MRAP1a, AAH62721.1; human MRAP1b, NP_996781.1), *Mus musculus* (mouse, NP_084120.1); *Macaca mulatta* (monkey, XP_001096328.3), *Gallus gallus* (chicken, XR_001470382.2), *Chrysemys picta bellii* (turtle, XP_005283970.1), *Xenopus tropicalis* (frog, XP_002938489.2), *Danio rerio* (zebrafish, ENSDART00000148193.3), *Sus scrofa* (pig, XP_020926573.1), *Cricetulus griseus* (chinese hamster, XP_003495626.1), *Rattus norvegicus* (rat, NP_001129306.1), *Lagenorhynchus obliquidens* (dolphin, XP_026957114.1), *Vulpes vulpes* (red fox, XP_025840964.1), *Delphinapterus leucas* (whale, XP_022408665.2), and *Capra hircus* (goat, XP_005674803.2).

**Figure 3 biomolecules-12-01608-f003:**
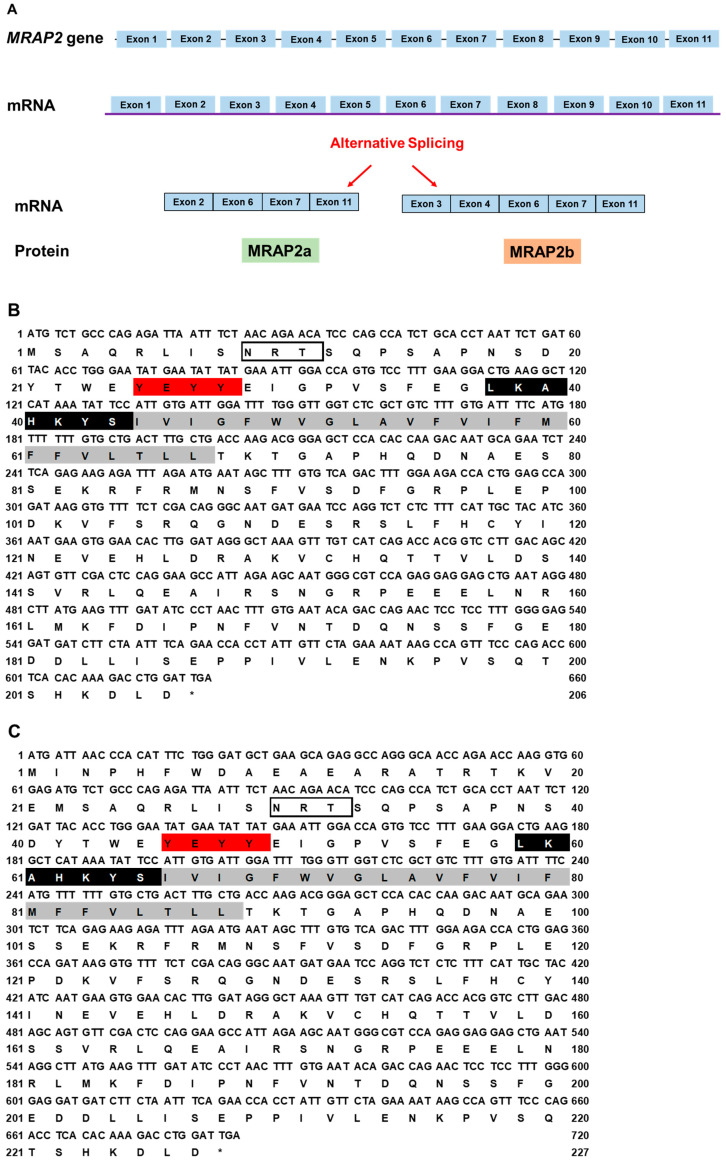
Schematic diagram (**A**), nucleotide and deduced amino acid sequences (**B**, cMRAP2a, **C**, cMRAP2b) and phylogenetic tree (**D**) of MRAP2s. Positions of nucleotide and amino acid sequences are indicated on both sides. *N*-linked glycosylation sites are present in open boxes. Red box denotes YEYY motif. Light blue box indicates LDYL motif. Black box is LKANKYL motif. Shaded box shows putative TMD. Asterisk (*) shows stop codon. The tree was constructed by the neighbor-joining (NJ) method. Numbers at nodes indicate the bootstrap value, as percentages, obtained for 1000 replicates. Black dot denotes canine MRAP1. MRAP1s: MRAP2s: *Canis lupus familiaris* (dog MRAP2a, XM_038682814.1; MRAP2b, XM_038682813.1), *Mus musculus* (mouse NP_001171202.1), *Sus scrofa* (pig, XP_003353296.2), *Capra hircus* (goat, XP_017908670.1), *Callorhinchus milii* (elephant shark, XP_007906624.1), *Balaenoptera musculus* (whale, XP_036727732.1), *Bos taurus* (bovine, NP_001092863.1), *Danio rerio* (zebrafish, MRAP2a: F8W4H9.1, MRAP2b: F8W4H9.1), *Gallus gallus* (chicken, ALO81626.1), *Mus caroli* (Ryukyu mouse, XP_021029091.1), and *Homo sapiens* (human, AAH10003.2).

**Figure 4 biomolecules-12-01608-f004:**
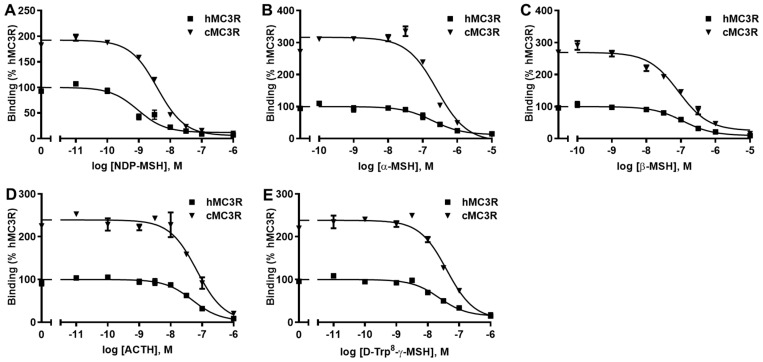
Ligand binding properties of cMC3R. Different concentrations of unlabeled NDP-MSH (**A**), α-MSH (**B**), β-MSH (**C**), ACTH (1-24) (**D**), and D-Trp^8^-γ-MSH (**E**) were used to displace the binding of ^125^I-NDP-MSH. Results are expressed as % of hMC3R binding ± range from duplicate determinations within one experiment. All experiments were repeated at least three independent times.

**Figure 5 biomolecules-12-01608-f005:**
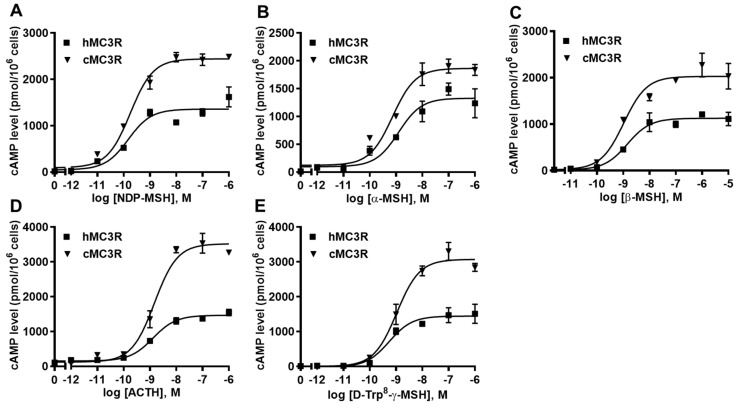
Signaling properties of cMC3R. HEK293T cells were transiently transfected with MC3R plasmids. Different concentrations of NDP-MSH (**A**), α-MSH (**B**), β-MSH (**C**), ACTH (1-24) (**D**), and D-Trp^8^-γ-MSH (**E**) were used to stimulate the cells. Data are means ± SEM from triplicate measurements within one experiment. All experiments were performed at least three times independently.

**Figure 6 biomolecules-12-01608-f006:**
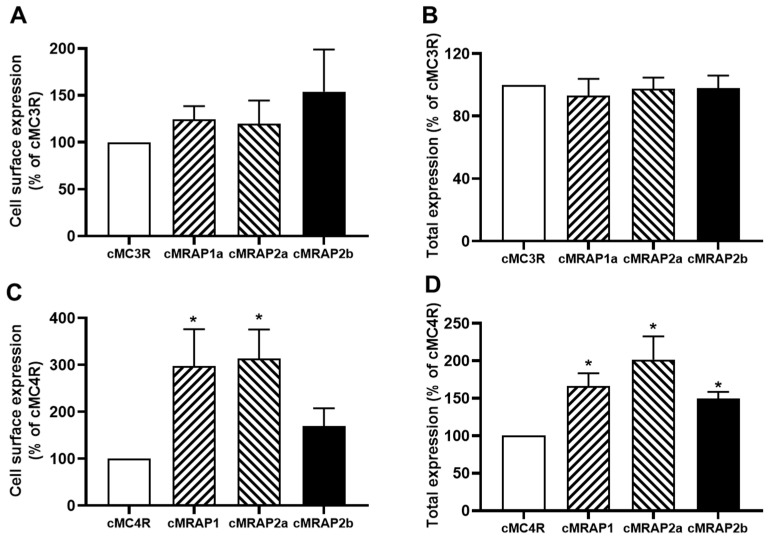
Regulation of cMC3R (**A**,**B**) and cMC4R (**C**,**D**) expression by MRAPs. Cell surface and total expression of cMC3R and MC4R was measured by flow cytometry. HEK293T cells were co-transfected with cMC3R or cMC4R and cMRAPs (1:5). Fluorescence in cells transfected with empty vector pcDNA3.1 was used for background staining. The results were calculated as % of 1:0 group. Each data point represented the mean ± SEM (n = 3–4). * Indicates significantly different from 1:0 group (*p* < 0.05) (one-way ANOVA followed by Tukey test).

**Figure 7 biomolecules-12-01608-f007:**
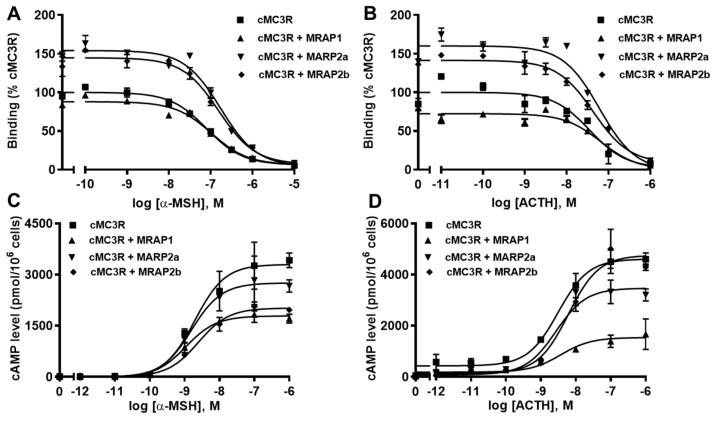
Modulation of cMC3R pharmacology by MRAPs. Ligand binding (**A**,**B**) and signaling (**C**,**D**) properties of cMC4R to α-MSH or ACTH (1-24) upon co-expression of cMC4R with cMRAP1, cMRAP2a or cMRAP2b were measured. Results of binding properties were calculated as % of cMC4R without MRAPs, from duplicate determinations within one experiment. All experiments were performed at least three independent times.

**Figure 8 biomolecules-12-01608-f008:**
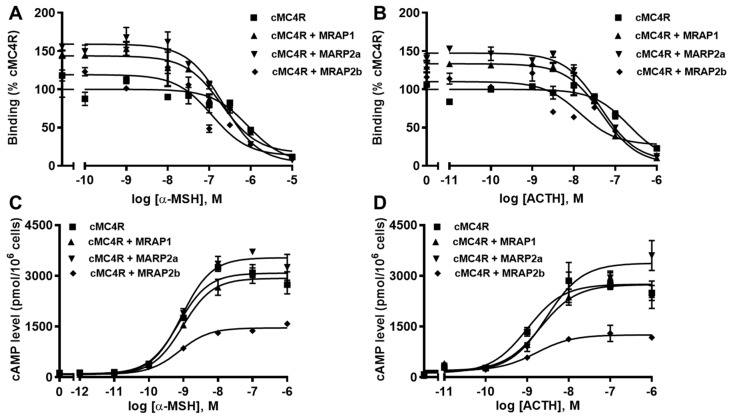
Modulation of cMC4R pharmacology by MRAPs. Ligand binding (**A**,**B**) and signaling (**C**,**D**) properties of cMC4R to α-MSH or ACTH (1-24) upon co-expression of cMC4R with cMRAP1, cMRAP2a or cMRAP2b were measured. Results of binding properties were calculated as % of cMC4R without MRAPs, from duplicate determinations within one experiment. All experiments were performed at least three independent times.

**Table 1 biomolecules-12-01608-t001:** The ligand binding properties of cMC3R.

MC3R	IC_50_ (nM)	cMC3R	hMC3R
B_max_ (%)		247.97 ± 13.44 ^b^	100
NDP-MSH	IC_50_ (nM)	4.37 ± 0.72 ^a^	1.98 ± 0.43
α-MSH	IC_50_ (nM)	240.42 ± 37.34	221.57 ± 30.23
β-MSH	IC_50_ (nM)	140.18 ± 31.37	161.56 ± 31.45
ACTH	IC_50_ (nM)	58.52 ± 8.67	45.71 ± 5.23
D-Trp^8^-γ-MSH	IC_50_ (nM)	39.30 ± 0.30 ^b^	24.62 ± 1.25

Values are expressed as the mean ± SEM of at least three independent experiments. ^a^ Significant difference from the parameter of hMC3R, *p* < 0.05. ^b^ Significant difference from the parameter of hMC3R, *p* < 0.001.

**Table 2 biomolecules-12-01608-t002:** The signaling properties of cMC3R.

MC3R	EC_50_/R_max_	cMC3R	hMC3R
Basal (%)		100.72 ± 7.69	100
NDP-MSH	EC_50_ (nM)	0.40 ± 0.16	0.24 ± 0.11
	R_max_ (%)	199.38 ± 28.06 ^a^	100
α-MSH	EC_50_ (nM)	0.97 ± 0.35	1.49 ± 0.22
	R_max_ (%)	145.69 ± 14.30	100
β-MSH	EC_50_ (nM)	1.04 ± 0.23	1.46 ± 0.55
	R_max_ (%)	229.77 ± 46.45 ^a^	100
ACTH	EC_50_ (nM)	1.46 ± 0.30	2.12 ± 0.69
	R_max_ (%)	182.83 ± 25.21 ^a^	100
D-Trp^8^-γ-MSH	EC_50_ (nM)	1.08 ± 0.15	0.71 ± 0.19
	R_max_ (%)	188.25 ± 19.56 ^a^	100

Values are expressed as the mean ± SEM of at least three independent experiments. ^a^ Significant difference from the parameter of hMC3R, *p* < 0.05.

**Table 3 biomolecules-12-01608-t003:** The effect of MRAPs on ligand binding properties of cMC3R.

cMC3R/cMRAPs	B_max_ (%)	α-MSH	ACTH
		IC_50_ (nM)	IC_50_ (nM)
cMC3R	100	106.68 ± 13.80	55.56 ± 11.64
cMC3R/cMRAP1	81.81 ± 5.01 ^a^	127.76 ± 30.47	68.59 ± 9.31
cMC3R/cMRAP2a	148.43 ± 14.94 ^a^	118.61 ± 22.25	57.58 ± 4.09
cMC3R/cMRAP2b	128.42 ± 9.06 ^a^	148.03 ± 30.16	49.59 ± 6.77

Values are expressed as the mean ± SEM of at least three independent experiments. ^a^ Significant difference from the parameter of cMC3R, *p* < 0.05.

**Table 4 biomolecules-12-01608-t004:** The effect of MRAPs on cAMP signaling of cMC3R.

cMC3R/cMRAPs	Basal (%)	α-MSH		ACTH	
		EC_50_ (nM)	R_max_ (%)	EC_50_ (nM)	R_max_ (%)
cMC3R	100	3.08 ± 1.26	100	2.02 ± 0.39	100
cMC3R/cMRAP1	100.91 ± 16.74	0.88 ± 0.18	46.67 ± 8.88 ^b^	2.11 ± 0.56	46.61 ± 12.37 ^a^
cMC3R/cMRAP2a	84.71 ± 19.03	1.37 ± 0.26	73.02 ± 5.81 ^a^	2.18 ± 0.39	73.79 ± 4.94 ^a^
cMC3R/cMRAP2b	68.88 ± 9.35 ^a^	3.64 ± 1.11	66.93 ± 9.64 ^a^	3.41 ± 0.99	85.02 ± 11.04

Values are expressed as the mean ± SEM of at least three independent experiments. ^a^ Significant difference from the parameter of cMC3R, *p* < 0.05. ^b^ Significant difference from the parameter of cMC3R, *p* < 0.001.

**Table 5 biomolecules-12-01608-t005:** The effect of MRAPs on ligand binding properties of cMC4R.

cMC4R/cMRAPs	B_max_ (%)	α-MSH	ACTH
		IC_50_ (nM)	IC_50_ (nM)
cMC4R	100	504.66 ± 124.33	348.29 ± 94.51
cMC4R/cMRAP1	151.36 ± 10.67 ^a^	186.62 ± 13.60 ^a^	37.63 ± 6.84 ^b^
cMC4R/cMRAP2a	167.03 ± 11.83 ^c^	228.40 ± 40.74 ^a^	35.63 ± 8.10 ^b^
cMC4R/cMRAP2b	121.60 ± 10.06	120.19 ± 22.41 ^b^	16.93 ± 5.69 ^c^

Values are expressed as the mean ± SEM of at least three independent experiments. ^a^ Significant difference from the parameter of cMC4R, *p* < 0.05. ^b^ Significant difference from the parameter of cMC4R, *p* < 0.01. ^c^ Significant difference from the parameter of cMC4R, *p* < 0.001.

**Table 6 biomolecules-12-01608-t006:** The effect of MRAPs on signaling properties of cMC4R.

cMC4R/cMRAPs	Basal (%)	α-MSH		ACTH	
		EC_50_ (nM)	R_max_ (%)	EC_50_ (nM)	R_max_ (%)
cMC4R	100	1.18 ± 0.21	100	0.92 ± 0.41	100
cMC4R/cMRAP1	53.89 ± 4.06 ^b^	1.37 ± 0.24	99.80 ± 3.80	1.92 ± 0.40	76.21 ± 13.29
cMC4R/cMRAP2a	74.70 ± 4.28 ^b^	0.90 ± 0.10	87.00 ± 10.28	3.82 ± 1.25	144.41 ± 13.36 ^a^
cMC4R/cMRAP2b	55.82 ± 3.43 ^b^	0.63 ± 0.10	52.62 ± 7.93 ^b^	1.20 ± 0.44	55.87 ± 9.82 ^a^

Values are expressed as the mean ± SEM of at least three independent experiments. ^a^ Significant difference from the parameter of cMC4R, *p* < 0.05. ^b^ Significant difference from the parameter of cMC3R, *p* < 0.001.

## Data Availability

The raw data supporting the conclusions of this article will be made available by the authors upon request, without undue reservation.

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
