# Peer review of "Modulation of Canine Melanocortin-3 and -4 Receptors by Melanocortin-2 Receptor Accessory Protein 1 and 2"

_biomolecules, 2022, doi:10.3390/biom12111608_

Round 1

Reviewer 1 Report

In this study the authors present new information on the pharmacological properties of canine Mc3r and how co-expression with canine Mrap1 or the two canine Mrap2 slice variants affects ligand binding and ligand activation of the receptor.  The Introduction does provide the reader with the appropriate background information to understand the rationale for the study. The Methods section is organized in a logical manner, and the authors provide references for the standard procedures that they perform in this study without providing actual details of the procedures. I appreciated this economy of space. The Results are also organized in a logical manner and the Figures and Tables are very clear. I did find the Discussion a bit hard to follow due to the fact that the authors attempted to compare and contrast pharmacological studies for Mc3r and Mc4r on a very broad spectrum of species (many fish species which I believe were mostly teleosts).  By the end of the Discussion it was difficult to see a "big picture" with respect to phylogenetic trends in the pharmacology of Mc3r and Mc4r, or perhaps insights into the evolution of these receptors with respect to their apparent interactions with the Mraps. The Conclusion refocused on the canine receptor data which was helpful.

However, a question I could not ascertain from this study was whether Mc3r and Mc4r are co-expressed in canine neurons with these Mraps. Based on previous studies on other species, I expect that one or both of the Mrap2s are present in canine neurons that also express at least Mc4r. As for Mrap1, in some species this accessory does not appear to  be expressed in the CNS. What is the situation for the canine CNS? My point is I think this study will benefit from some discussion of the tissue distribution of the Mraps in the canine CNS and is there overlaps with the tissue distribution pattern for canine Mc3r and Mc4r. Hence, I feel that the study is incomplete at this stage.  

Minor points:

Line 15. Please re-write this sentence. "At MC3R" is an strange way to begin this sentence.

Line 18: This sentence also should be re-written.

Line 102 Please give the molar concentration for the receptor used in the reaction rather than the ug amount.

Author Response

Reviewer #1

Comment 1. I did find the Discussion a bit hard to follow due to the fact that the authors attempted to compare and contrast pharmacological studies for Mc3r and Mc4r on a very broad spectrum of species (many fish species which I believe were mostly teleosts). 

Reply: We tried to summarize all publications about pharmacological studies for MC3R and MC4R of different species, including mammals (human, pig, and panda), avian (chicken), amphibians (Mexican axolotl), and teleosts (zebrafish, grouper, Nile tilapia, topmouth culter, channel catfish, and snakehead). These studies and current results showed that MRAP1 and MRAP2 modulation of MC3R/MC4R trafficking, ligand binding, and signaling are species-dependent. We also revised manuscript to facilitate better reading and understanding.

Comment 2. By the end of the Discussion it was difficult to see a "big picture" with respect to phylogenetic trends in the pharmacology of Mc3r and Mc4r, or perhaps insights into the evolution of these receptors with respect to their apparent interactions with the Mraps. The Conclusion refocused on the canine receptor data which was helpful.

Reply: Thanks for your kind suggestions. We have revised the corresponding statement focused on canine receptor.

Comment 3. However, a question I could not ascertain from this study was whether Mc3r and Mc4r are co-expressed in canine neurons with these Mraps. Based on previous studies on other species, I expect that one or both of the Mrap2s are present in canine neurons that also express at least Mc4r. As for Mrap1, in some species this accessory does not appear to be expressed in the CNS. What is the situation for the canine CNS? My point is I think this study will benefit from some discussion of the tissue distribution of the Mraps in the canine CNS and is there overlaps with the tissue distribution pattern for canine Mc3r and Mc4r. Hence, I feel that the study is incomplete at this stage.  

Reply: This is an excellent suggestion. To our knowledge, MRAP2 is expressed in CNS from mammals to teleosts. As for MRAP1, although human MRAP1a and MRAP1b are primarily expressed in adrenal gland, MRAP1b has high expression in brain (Nat. Genet. PMID: 15654338). In other species, mrap1 also is detected in brain of Xenopus (J. Cell Physiol. PMID: 33521982) and chicken (J Endocrinol. PMID: 28512117).

We do not have access to canine tissues. Therefore, we cannot perform these experiments. Studies from scientists with access to canine tissue mRNA can provide this missing information and advance the field further. Furthermore, we have revised the corresponding statement using ‘potential MRAP regulation of MC3R or MC4R’ in the revised manuscript.

Comment 4. Line 15. Please re-write this sentence. "At MC3R" is an strange way to begin this sentence.

Reply: Done.

Comment 5. Line 18: This sentence also should be re-written.

Reply: Done.

Comment 6. Line 102 Please give the molar concentration for the receptor used in the reaction rather than the ug amount.

Reply: I do not agree with you on this comment. We published numerous manuscripts using ug amount for transfecting plasmid concentration. I also searched many publications in Science, Cell, and other high impact journals, almost all used ug amount, not molar unit, for plasmid concentration. Please see Science 2013, (PMID: 23869016), Cell, 2019 (PMID: 31002796), J. Med. Chem. 2022 (PMID: 35404053) for examples.

Reviewer 2 Report

The study seems interesting, and the authors have already published related research. In the present study, I could not find some novel results as the bioinformatics part included does not provide enough information. The phylogenetic analysis is confusing for me like why the canine sequence diverges so much from the counterparts and what is needed for the inclusion of so many fish, if it was necessary then the authors should discuss the results in detail. The discussion compares the data rather than explaining the current results. Authors have also given the figure numbers in the discussion which is not necessary unless it is in the journal style. The paper could be accepted if some more explanations regarding the present results are included.

Author Response

Reviewer #2

Comment 1. The study seems interesting, and the authors have already published related research. In the present study, I could not find some novel results as the bioinformatics part included does not provide enough information. The phylogenetic analysis is confusing for me like why the canine sequence diverges so much from the counterparts and what is needed for the inclusion of so many fish, if it was necessary then the authors should discuss the results in detail.

Reply: Our lab focuses on MC3R/MC4R pharmacology and regulation by MRAPs in different species, and have published several papers. The current and previous studies found that MRAPs modulate the MC3R/MC4R trafficking and pharmacology in species- and receptor-specific manner. The effects of MRAP1 and MRAP2 on canine MC3R and MC4R have not been studied before.

Comment 2. The discussion compares the data rather than explaining the current results.

Reply: Thanks for your kind suggestions. We have revised the manuscript to explain the current results (Line 322-329, Line 355-360, and Line 374-380).

Comment 3. Authors have also given the figure numbers in the discussion which is not necessary unless it is in the journal style.

Reply: Done.

Round 2

Reviewer 2 Report

The authors have included the corrections, so it can be accepted for publication.